# Novel insights into immune stress markers associated with myxosporeans gill infection in Nile tilapia (molecular and immunohistochemical studies)

Reem M. Ramadan[1], Olfat A. Mahdy[1], Mohamed A. El-Saied[2], Faten F. Mohammed[2,3]*, Mai A. Salem[1]

1 Faculty of Veterinary Medicine, Department of Parasitology, Cairo University, Giza, Egypt, 2 Faculty of Veterinary Medicine, Department of Pathology, Cairo University, Giza, Egypt, 3 Department of Pathology, College of Veterinary Medicine, King Faisal University, Al-Ahsa, Saudi Arabia

* fhanafi@kfu.edu.sa, fatenfathy21@yahoo.com

## Abstract

Nile tilapia (*Oreochromis niloticus*) is valued in aquaculture because of its quick development and ability to thrive in various environments. Myxosporeans are among the fish parasites that affect fish productivity, as they impact fish growth and reproduction, resulting in large fish deaths in farms and hatcheries. This study has been focused on morpho-molecular identification for the myxosporean parasites infecting Nile tilapia from three governorates in Egypt and assessment of gene expression of different cytokines (Interleukin-1βeta (*IL-1β*), major histocompatibility complex class II (*MHC-II*), and clusters of differentiation 4 (*CD-4*) and 8 (*CD-8*)) in tissues. Additionally, this work aimed to correlate the developed histopathological alterations and inflammatory reactions in gills with immunohistochemical expression of inducible nitric oxide synthase (*iNOS*) and tumor necrosis factor-alpha (*TNF-α*). Finally, the infected fish's cortisol levels and blood glucose were assessed. Results of BLAST sequence analysis of the 18S rRNA for the collected protozoans confirmed *Myxobolus agolus*, *M. brachysporus*, *M. tilapiae*, and *Henneguya* species. The molecular characterization of the immunological status of gills revealed marked upregulation of different inflammatory cytokines in the gills of infected fish. There was a significantly increased serum cortisol and glucose level in infected fish compared with control, non-infected ones. Severe histopathological alterations were observed in the infected fish gills, associated with increased expression of *iNOS* and *TNF-α* and related to myxosporean infection. The present study provides new insights into oxidative stress biomarkers in Nile tilapia infected with Myxosporeans and elucidates the gill's immune status changes as a portal of entry for protozoa that contribute to tissue damage.

## Introduction

Tilapia species, especially *Oreochromis niloticus (O. niloticus)*, generally known as Nile tilapia, are highly valued in aquaculture due to their rapid growth and ability to grow under various

**Data Availability Statement:** All the authors declare that all relevant data are within the

manuscript. The datasets generated or analyzed during the current study are available in the GENE BANK repository, accession numbers: [PP318831, PP319004, PP309819 and PP317555].

**Funding:** Funding: This study was supported by the Deanship of Scientific Research, Vice Presidency for Graduate Studies and Scientific Research at King Faisal University, Saudi Arabia, for financial support under the annual funding track GRANT5,913. The funder had no role in study design, data collection and analysis, decision to publish, or preparation of the manuscript. There was no additional external funding received for this study.

**Competing interests:** The authors have declared that no competing interests exist.

environmental circumstances [1]. Due to the rising demand for fish proteins, emphasis is being directed to parasites impacting *O. niloticus* growth, health, and survival, as illnesses significantly limit fish productivity [2]. Myxosporeans are among the fish parasites that affect fish productivity. They impact fish growth and reproduction and can cause epizootics, resulting in significant fish deaths on farms and hatcheries. Myxosporeans can harm the health of immunocompromised people who eat infected fish [3].

Molecular analyses are becoming a crucial tool in the study of myxosporean parasites, helping to confirm the identification process and being widely applied to the taxonomy of these parasites, as well as determining the phylogenetic identification of myxosporeans inside metazoans when combined with traditional techniques like precise location of sporulation, spore morphology, and host and tissue specificity. Furthermore, this has resulted in accurately identifying and distinguishing morphologically identical myxobolid species (sp.) from the genera *Henneguya* and *Myxobolus* [4].

*Myxobolus* infection increases cell immunity and generates interleukins and proinflammatory cytokines. These interleukins stimulate mast cells and eosinophils, which raise blood levels of immunoglobulin E (IgE) and immunoglobulin G (IgG) [4]. Proinflammatory cytokines such as interleukin are generated during the early phases of infection and play a crucial role in the pathogenesis of parasitic infections. According to reports of infections, *IL-1β* genes are essential in limiting the growth of parasites and producing a chronic latent infection through cyst formation [5]. Immune cells stimulated by various pathogens (including bacterial, viral, or parasitic components) release cytokines. They can alter immune responses to eliminate pathogens and attract neutrophils, lymphocytes, and macrophages to the affected areas of innate immunity when they bind to the right receptors [6].

Major histocompatibility complex (*MHC*) genes have garnered significant attention in evolutionary biology due to their pivotal function in facilitating parasite resistance, as noted by Sommer and Schad et al. [7,8]. The primary role of *MHC* class II (*MHC-II*) genes is to encode glycoproteins on the cell surface, which bind and deliver foreign peptides to the T-cell receptor. Expression of the mutually exclusive cluster of differentiation 4 or 8 (*CD-4* or *CD-8*) coreceptor distinguishes functionally distinct lymphocytes from one another [6,9].

This study aimed to provide morphological descriptions and phylogenetic analysis depending on small subunit rDNA sequence data for the detected myxosporean parasites. However, no research in Egypt has been conducted on fish's immunological status and associated cytokines during infection with *Myxobolus* and *Henneguya* species. This study used quantitative real-time polymerase chain reaction (qRT-PCR) to analyze cytokine gene expression in Nile tilapia gills, specifically *IL1-β*, *MHC-II*, *CD-4*, and *CD-8*. Furthermore, affected fish's cortisol levels and blood glucose were assessed as a physiological response to parasite protozoan infection. Additionally, a correlation was performed between the developed histopathological alterations and inflammatory reactions in gills and the immunohistochemical expression of *iNOS* and *TNF-α*.

## Material and methods

### Ethical approval

The current study has been approved by the Cairo University Faculty of Veterinary Medicine's Institutional Animal Care and Use Ethical Committee (IACUC-Vet-25122023810).

### Fish sample collection and parasitological investigation

A total of 628 specimens of *O. niloticus*, commonly known as the Nile tilapia, were randomly sampled from freshwater habitats between February and December 2023. Sampling locations

included different points along the Nile River within the governorates of Giza (348 specimens), Kafr Elsheikh (172 specimens), and Beni-Sueif (108 specimens).

The fish were housed alive in glass aquariums equipped with continuous aeration and filtration systems. They were supplied with tap water devoid of chlorine [10]. All fish were then immediately moved to the Laboratory of Parasitology at the Faculty of Veterinary Medicine, Cairo University, Egypt. The fish were anesthetized by immersion in an anesthetic bath containing 150 mg/L of tricaine methanesulfonate (MS-222), facilitating rapid absorption through the gills into the bloodstream. Subsequently, the fish are euthanized under anesthesia to alleviate their suffering. Euthanasia is carried out through decapitation. Fish samples were identified following Mahmoud et al. [11]. Fish samples were collected and examined to ascertain the presence or absence of mature myxospore stages. A comprehensive external examination was conducted to detect macroscopically visible lesions or cysts throughout the body. Standard procedures were followed to remove the gills, clean them, and check them under a microscope for myxosporean parasites. Impression smears emerged after the gills were fixed and stained with Giemsa's dye. Myxosporean cysts were gathered and ruptured to produce spores for light microscopy. The minimum and maximum spore measurement values were determined using micrometers (μm). The methods of Abdel-Gaber et al. [12] were followed in the measurement, classification, and description of spores.

### Genotyping identification of myxosporean parasites

**DNA extraction.** Myxospores were isolated from samples preserved in ethanol by centrifugation at 5000 ×g for 10 minutes. After removing the supernatant, the spore pellet was incubated for three to four hours at 55˚C. Subsequently, 500 μL of lysis STE buffer (10 mM Tris −HCl [pH 8], 100 mM NaCl, 10 mM ethylenediamine-tetraacetic acid [EDTA], 0.2% sodium dodecyl sulfate, and 0.4 mg proteinase K) was added. Genomic DNA was extracted using the QIAamp DNA MiniKit (Qiagen) following the manufacturer's instructions [13].

**PCR amplification.** Amplification of the 18S rRNA gene (small subunit ribosomal RNA) was used to identify myxosporean parasites from fish samples using conventional PCR, as described by Khalifa et al. [14]. A total volume of 25 μL was employed for the amplification reaction, consisting of 1 μL of 10 pmol of each primer, 3 μL of extracted DNA serving as a template, and 12.5 μL of Cosmo Taq DNA polymerase Master Mix (Willowfort, UK). Table 1 presents specific primer sequences and PCR conditions. The positive and negative controls used the same protocols: genomic DNA from well-known protozoan parasites and nuclease-free water.

**Table 1. Oligonucleotide primer pairs used in PCR amplifications.**

| Parasite | Target Gene | Primer sequence | Amplification conditions | Reference |
|---|---|---|---|---|
| *Myxobolus sp.* | 18S rRNA | F: 5' CTGCGGACGGCTCAGTAAATCAGT 3'<br>R: 5' CCAGGACATCTTAGGGCATCACAG 3' | 35 cycles of 94˚C for 50 s, 56˚C for 50 s, 72˚C for 80 s, and followed by 72˚C for 7 min | [12] |
| | | F: 5' CCTGAGAAACGGCTACCACATCCA 3'<br>R: 5' GATTAGCCTGACAGATCACTCCACGA 3' | | |
| | | F: 5' GATGATTAACAGGAGCGGTTGG 3'<br>R:5' ACCGCTCCTGTTAATCATCACC 3' | | |
| *Henneguya sp.* | | F: 5' TTCTGCCCTATCAACTWGTTG 3'<br>R: 5' GGTTTCNCDGRGGGMCCAAC3' | 30 cycles of 95 ˚C for 1 min, 48 ˚C for 1 min and 72 ˚C for 2 min, followed by 10 min incubation at 72 ˚C | [15] |

F: Forward, R: Reverse, min: Minutes, s: Second.

**Sequencing and phylogenetic study.** The QIAquick purification extraction kit (Qiagen, Hombrechtikon, Switzerland) was utilized to purify the PCR products before they were sequenced utilizing the BigDye terminator V3.1 sequencing kit (Applied Biosystems, Waltham, MA, USA) [16]. Concurrent construction of the sequences was performed using the ChromasPro program (ChromasPro 1.7, Technelysium Pty Ltd., Tewantin, Australia). Following their BLAST match to GenBank reference sequences, they were examined using MEGA program 11. Phylogenetic trees were generated with 1000 bootstrap replicates using the neighbor-joining method and MEGA 11 software [17].

## Immunological statue of cytokine gene expressions in Nile tilapia by qRT-PCR

**Sampling.** Under completely sanitary conditions, ten highly infected fish with *Myxobolus* and *Henneguya* sp. were aseptically dissected to get the infected gills. Five unaffected fish were sampled to serve as negative controls. The gene of β-actin was utilized as a reference gene to normalize the expression of the *IL-1β*, *MHC-II*, *CD-4*, and *CD-8* genes. Infected fish tissues were finally preserved aseptically to isolate RNA and kept at -20°C [18].

**RNA isolation.** Total RNAs were obtained from gill samples stored at -80°C using the purification kit (Jena Bioscience, Germany) for total RNA according to the manufacturer's instructions. DEPC water containing 40 μL was used to elute the RNA. The ND-1000 Nano-Drop spectrophotometer, manufactured by Thermo Scientific, located in Waltham, Massachusetts, USA, was utilized to assess the integrity of the eluted RNA and ascertain its quantity. To prepare RNA samples for first-strand cDNA synthesis, residual genomic DNA was eliminated using the following procedure: A 30-minute incubation period at 37°C was initiated after combining the following components in an RNase-free tube: 10× reaction buffer containing $MgCl_2$, 1 μg of RNA sample, 10 μL of DEPC-treated water, and 1 μL of DNase I. The mixture was then returned to 65°C for 10 minutes, and 1 μL of 50 mM EDTA was added. Subsequently, the RevertAid First-Strand cDNA synthesis kit (Thermo Scientific, Waltham, USA) was used to produce the second strand cDNA by reverse transcription, utilizing the resultant RNA as a template. The cDNA was kept on ice and kept at -20°C, according to Mahdy et al. [18].

**The qRT-PCR protocol.** Using reference genes (β-actins) that were generated based on known sequences, four cytokines (*IL-1β*, *MHC-II*, *CD-4*, and *CD-8*) were assessed (Table 2). Applied BiosystemsTM 7500 Real-Time PCR Systems (Applied Biosystems, Bedford, MA, USA) were used to quantify cytokine expression levels in infected pigeons. The following items made up the qPCR system: 0.5 μL of each forward and reverse primer (10 pmol/L), 5 μL of ddH2O up to 10 μL, 0.2 μL ROX Dye (50×), and the WizPureTM qPCR Master (SYBR)

**Table 2. Pair primers used in the qRT-PCR.**

| Primers | Sequence (5′–3′) | References |
|---|---|---|
| *IL-1β* | F-TGGCCCTGACTGAACCACTG<br>R-TCAGACCCACGCCACAGAAC | [20] |
| *MHC-II* | F-ATGTCCAAGCTGCTGAAGATT<br>R-TGCCGTCTGACTTCTTCACC | [21] |
| *CD-4* | F-AAGAAACAGATGCGGGAGAGT<br>R-AGCAGAGGGAACGACAGAGAC | [22] |
| *CD-8* | F-ACACCAATGACCACAACCATAGAG<br>R-GGGTCCACCTTTCCCACTTT | [23] |
| *β-actin* | F-GGCTACTCCTTCACCACCACAG<br>R-GGGCAACGGAACCTCTCATT | |

(Wizbiosolution, Republic of Korea). For the qPCR reaction, the following conditions were followed: annealing for one minute at 58˚C, denaturation for 40 cycles at 95˚C for 15 s each, and predenaturation for 10 minutes at 95˚C, with a melting-c program set at 95˚C for 15 s, 60˚C for 1 minute, 95˚C for 30 s, and 60˚C for 15 s, the product specificity has been calculated. According to Mahdy et al. [19]., every experiment was conducted thrice.

**Measurements of cortisol and glucose levels.** Blood samples from five infected fish and five uninfected fish were subjected to biochemical analysis. Fish treated with benzocaine (50 mg/L) as an anesthetic, both with and without the use of an anticoagulant (heparin), had blood drawn from their caudal veins. The sample was centrifuged for 15 minutes at 1500 rpm after allowing it to coagulate at 4˚C to extract serum by biochemical analysis. Serum samples were then frozen at -20˚C. A commercial kit from Egypt's Spectrum Diagnostics and a Tokyo, Japan-based JASCO V-730 spectrophotometer were used to measure the glucose levels. A liquid scintillation counter was used to quantify radioactivity, and a commercial cortisol kit was used for radioimmunoassay to measure blood cortisol levels [24].

**Histological and immunohistochemical examination.** Representative pooled samples from the gills of heavily infected fish were adequately preserved in 10% neutral buffered formalin. Subsequently, they were sectioned at 4 μm thickness, embedded in paraffin blocks, cleared in xylene, and routinely stained with hematoxylin and eosin (H&E) stain [25].

Adhesive slides with tissue slices at 5 μm were prepared for immunostaining. Phosphate-buffered saline was used to rehydrate and rinse the sections. Primary monoclonal antibodies were diluted 1:200 and incubated overnight at 4˚C in a humid chamber against *TNF-α* (sc-52746; Santa Cruz Biotechnology, Inc., Heidelberg, Germany) and *iNOS* (sc-7271; Santa Cruz Biotechnology, Inc., Heidelberg, Germany). Hydrogen peroxide was used to block endogenous peroxidases. The positive reaction was then developed by following the manufacturer's instructions and using the HRP-labeled detection kit (Bio SB, USA). The process of eliminating the primary antibody incubation yielded negative controls. Tissue sections were examined using an Olympus BX43 light microscope and captured by an Olympus DP27 camera linked to CellSens Dimension software.

**Statistical analysis.** Protozoa measures were provided as means ± standard deviation. All statistical inference was performed using the PASW Statistics, ANOVA test (P-value $\leq 0.001$), Version 27 program (SPSS Inc., Chicago, IL, USA) [26–28].

## Results

### Prevalence of the detected protozoa among the inspected Nile tilapia

Macro and microscopic examination revealed that 194 of 628 (30.9%) collected *O. niloticus* were naturally infected with *Myxobolus* sp. (22.5%) and *Henneguya* sp. (8.4%) (Table 3). Giza governorate showed the highest level of infection, followed by Kafr-Elsheikh and Beni-Sueif governorates in both *Myxobolus* and *Henneguya* species. No mixed infections were found. All

**Table 3. Prevalence of the detected protozoa among inspected Nile tilapia.**

| Governorate | No. of examined | No. of infected (%) | Protozoon sp. | |
|---|---|---|---|---|
| | | | *Myxobolus* sp. (%) | *Henneguya* sp. (%) |
| Giza | 348 | 119 (34.2) | 96 (27.6) | 23 (6.6) |
| Kafr-Elsheikh | 172 | 51 (29.7) | 32 (18.6) | 19 (11.05) |
| Beni-Sueif | 108 | 24 (22.2) | 13 (12.04) | 11 (10.2) |
| Total | 628 | 194 (30.9) | 141 (22.5) | 53 (8.4) |

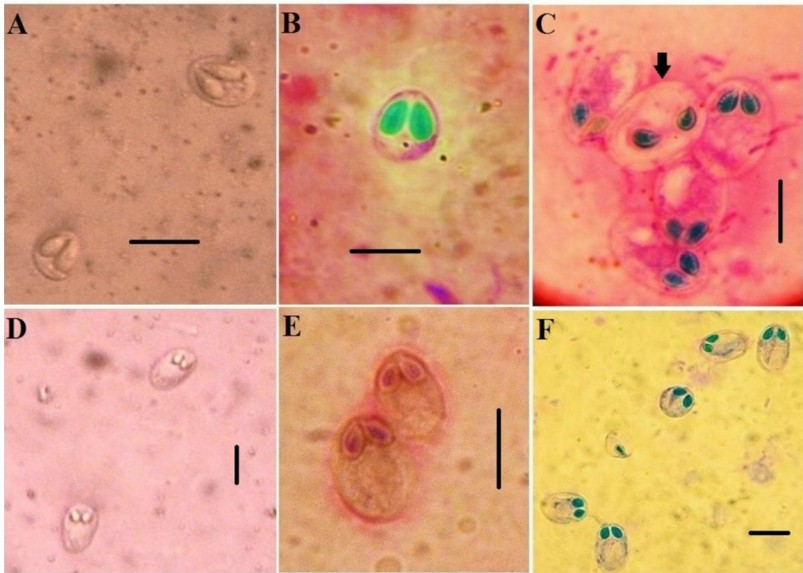

**Fig 1. Photomicrographs showing fresh and stained spores of *Myxobolus* sp. Giemsa-stained.** A) Fresh specimens of *M. agolus*. B) Giemsa-stained *M. agolus*. C) Giemsa-stained *Myxobolus brachysorus*. D-E) Fresh specimens of *M. tilapiae*. F) Giemsa-stained *M. tilapiae*. Scale bar = 10 μm.

recovered mature *Myxobolus* parasites belonged to 3 different sp., namely *M. brachysporus*, *M. agolus*, and *M. tilapiae*.

## Morphological identification

**Myxobolus agolus.** The spores had a rounded and slightly wider posterior end and were oval to slightly pyriform. Their length was 8.9 ± 12.1 (10.4 ± 1.5) μm, and their breadth was 8.2 ± 9.6 (8.9 ± 0.7) μm. The spore's anterior end was adorned with two pyriform, elongated polar capsules that hung for about two-thirds of its length. Along the inner wall of the capsule, the polar filament is coiled into eight or nine coils. The final third of the spore chamber was filled with sporoplasm. A sizable, spherical iodinophilous vacuole was observed in the sporoplasm (Fig 1A and 1B).

**Myxobolus brachysporus.** The ellipsoidal spores were significantly larger than the 6.9–8.2 (7.5 ± 0.7) μm in length, measuring 12.1–13.9 (13 ± 0.8) μm in width. The polar capsules primarily had a subcircular shape and were uniform in size. The polar filament coils in five to six turns, and the capsule's longitudinal axis is either slightly inclined or perpendicular to it. Sporoboplasm filled the remaining space inside the spore cavity. The sporoplasm had a large, spherical, iodinophilous vacuole (Fig 1C).

**Myxobolus tilapiae.** The spores were ellipsoidal to oval, with rounded ends on both the anterior and posterior sides. The posterior end appeared to be slightly wider than the anterior end. The spores' measurements were 15.2–16 (15.6 ± 0.4) μm in length and 9.7–10.9 (10.3 ± 0.6) μm in width. The spore's two oval, slightly asymmetrical polar capsules were in front. They accounted for over a quarter of the spore's length. The polar filaments were coiled securely within the polar capsules in five to six rotations after being positioned obliquely to the longitudinal axis of the polar capsule. The remaining spore space behind the polar capsules was occupied by binucleated sporoplasm with a rounded iodinophilous vacuole (Fig 1D–1F).

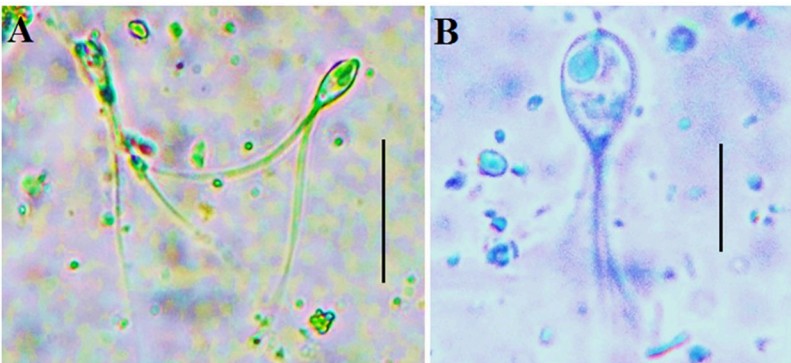

**Fig 2.** (A-B) *Henneguya sp*. from Nile tilapia, two posterior flagella, and two polar capsules are indicated. Scale bar = 10 μm.

**Henneguya sp.** Spores with a large oval body measuring (10–14) x (6.5–8) μm and two bifurcated filaments measuring approximately (41–53) μm in length were found in the tails of the spores. The equal-sized pyriform polar capsules, measuring approximately (2.9–4.2 x 1.6–2.4) μm in size, were situated at the anterior end. The number of polar filament coils is 3–5 in fresh spores. Seventy-five percent of the spore cavity was filled with granulated sporoplasm, which included two distinct sporoplasmic nuclei and one iodinophilous vacuole (Fig 2).

## Molecular identification

To accurately identify *Myxobolus* and *Henneguya* species isolated from Nile tilapia, morphologically identified myxosporean parasites were subjected to PCR amplification. Subsequently, the obtained products underwent DNA sequencing. Based on BLAST sequence analysis, the positive PCR results of the 18S rRNA gene were categorized into two genera: *Myxobolus* and *Henneguya* sp. (Figs 3 and 4). Pair-wise sequence alignment showed that sequences of *Myxobolus* sp. were confirmed as *M. agolus, M. brachysporus, and M. tilapiae* (Fig 3). Table 4 lists the accession numbers of the 18S rRNA gene of *Myxobolus* and *Henneguya* sp. deposited in GenBank.

## Immunological gene expression in infected Nile tilapia with the investigated protozoa

Infected tilapia had higher *IL-1β, MHC-II, CD-4,* and *CD-8* expression levels than control tilapia (Fig 5).

## Levels of stress indicator enzymes with glucose and cortisol

The cortisol and glucose levels were significantly higher in infected fish than in control, non-infected ones (Table 5). There is a significant difference between infected and non-infected fish with a P-value ≤ 0.001.

## Histopathological alterations

The microscopic examination of the gills revealed an intense inflammatory reaction involving the entire gill structure. The gill filaments showed severe congestion of the lamellar venous sinuses associated with hemorrhages (Fig 6A). In addition, marked distortion of gill filaments

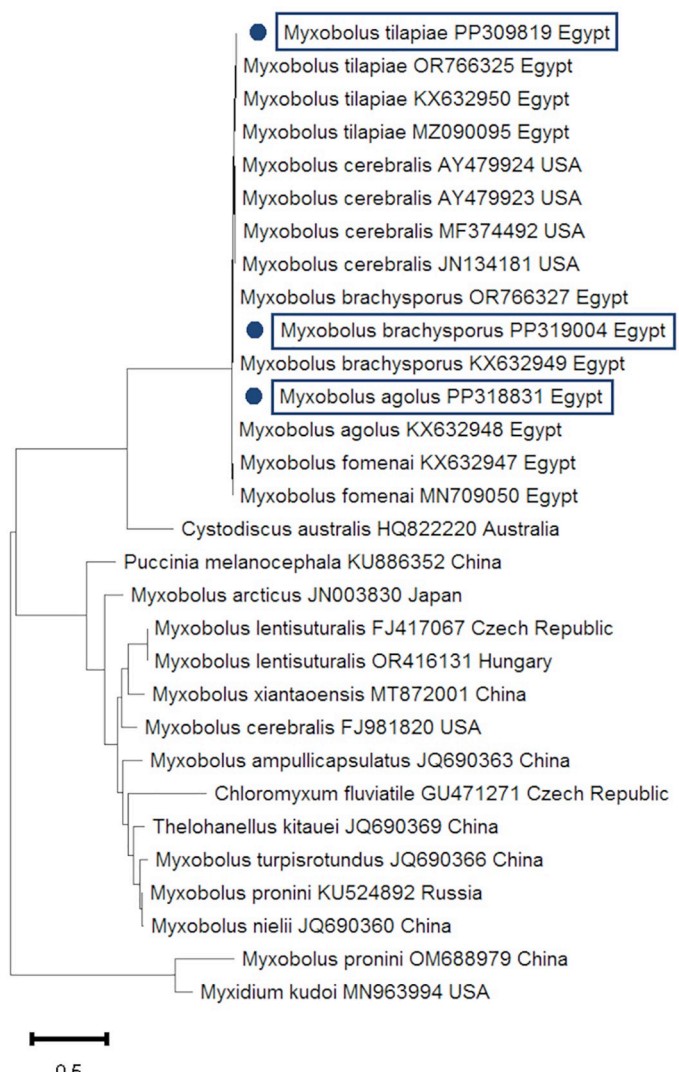

**Fig 3. A phylogenetic tree constructed using the neighbor-joining method on the 18S rRNA of *Myxobolus* sp.** For this query, the accession numbers are indicated by blue dots. The scale bar at the tree's base indicates how many nucleotide changes occur per site.

was observed. Moreover, severe hyperemia of the lamellar capillary sinuses of secondary gill lamellae associated with interlamellar hemorrhages was detected (Fig 6B), and there was extensive lamellar edema with sloughing of the lamellar epithelium admixed with leucocytes (Fig 6C).

The gill arch exhibited intense leucocytic infiltration with marked hyperemia of afferent and efferent branchial arteries and arterioles (Fig 6D). The inflammatory reaction primarily comprised lymphocytes and histiocytes, mixed with eosinophilic granular cells (EGC) (Fig 6E). In addition, different sporogenic stages of *Myxobolus* sp., mixed with lymphocytes and histiocytes, were detected in branchial arteries (Fig 6F and 6G), indicating the circulatory pathway of protozoal dissemination. The mucosal epithelium of the gill arch showed mucous cell hyperplasia with lymphocytic infiltration (Fig 6H).

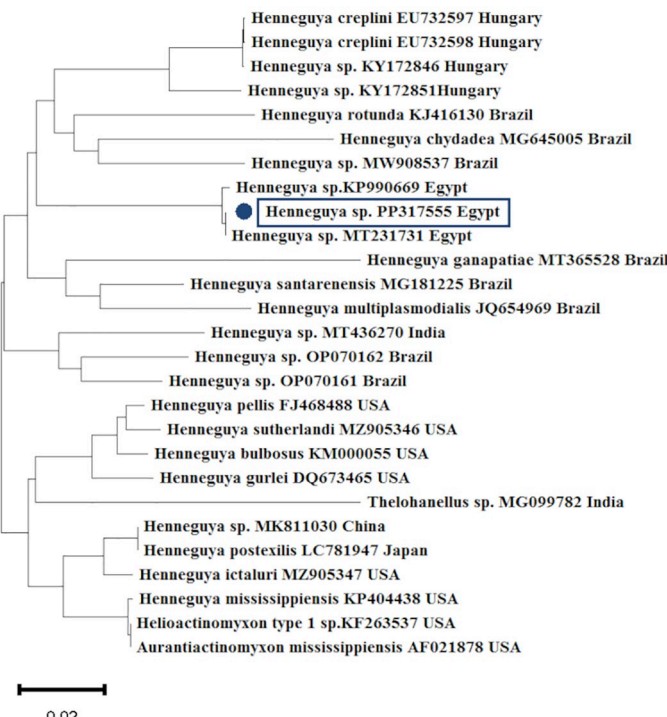

**Fig 4. A phylogenetic tree constructed using the neighbor-joining method on the 18S rRNA of *Henneguya* sp.** For this query, the accession numbers are indicated by blue dots. The scale bar at the tree's base indicates how many nucleotide changes occur per site.

There was a correlation between the existence of *Myxobolus* sp. in gills and the intensity of the inflammatory reaction. Various stages of *Myxobolus* sp. were observed in the gill arch, accompanied by a notable inflammatory response. The parasitic cysts within the gill arch contained different developmental stages of *Myxobolus* sp., characterized by distinctive paired polar capsules (Fig 7A and 7B).

### The expression of *TNF-α* and *iNOS* in gills

The microscopic examination of the gills revealed augmented expression of *iNOS* and *TNF-α* in the gills. The *iNOS* expression was observed in vascular smooth muscle cells of the vascular wall and mononuclear cells in the area of inflammatory reaction comprising gill structures, mainly the gill arch. while *TNF-α* expression was observed in mononuclear cell infiltration of the stroma of the gill arch (Fig 7C and 7D).

**Table 4. The accession numbers of *Myxobolus* and *Henneguya* sp.**

| Samples | Amplified products (Base pair) | Target gene | Accession number | % of identity |
|---|---|---|---|---|
| *M. agolus* | (1400) | 18S rRNA | PP318831 | KX632948 (100%) |
| *M. brachysporus* | (1339) | | PP319004 | KX632949 (100%); OR766327 (100%) |
| *M. tilapiae* | (1348) | | PP309819 | KX632950 (100%); OR766325 (100%) |
| *Henneguya sp.* | (876) | | PP317555 | MT231731 (100%); KP990669 (99.64%) |

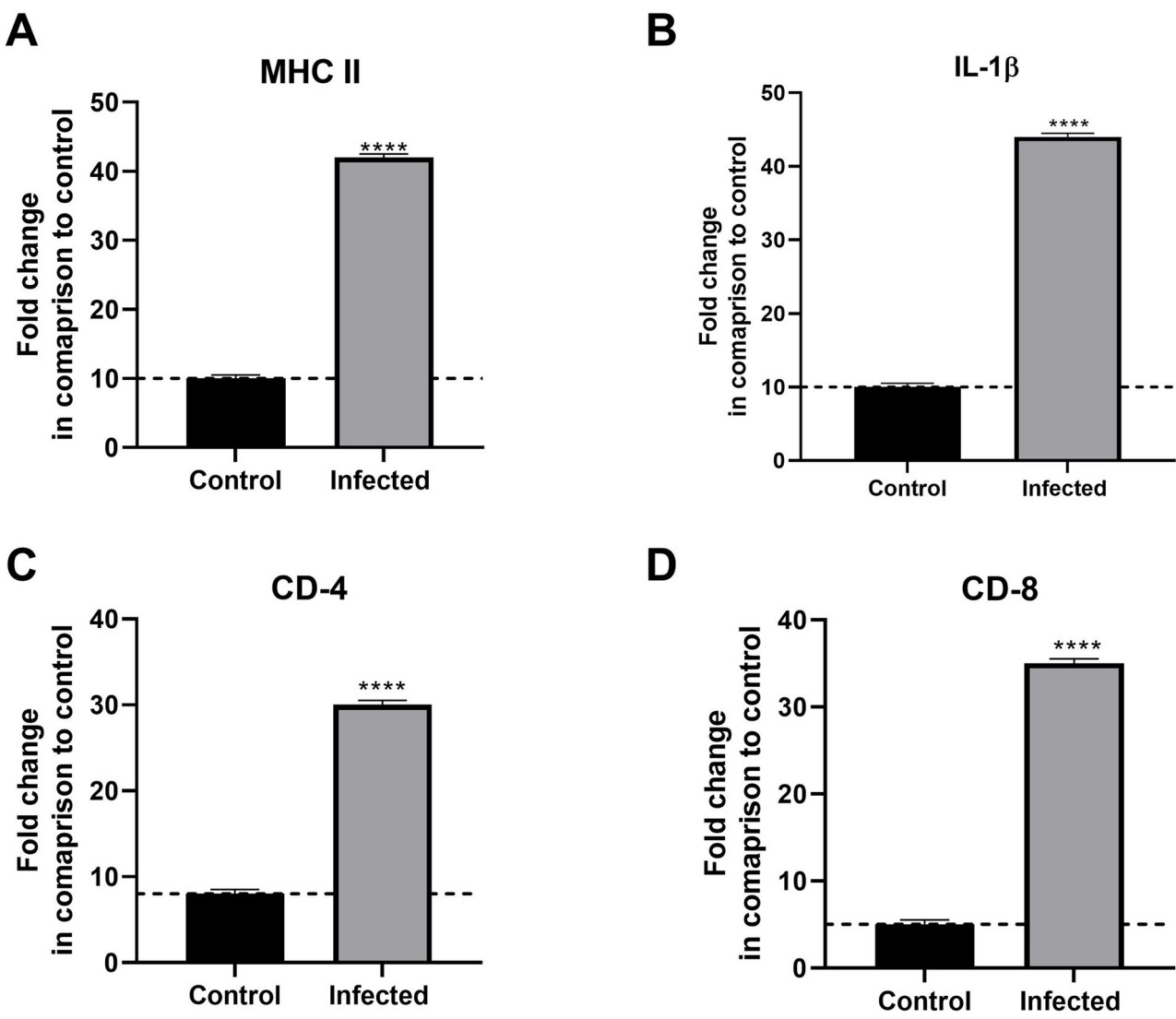

**Fig 5. Gene expression of *MHC-II*, *IL-1β*, *CD-4*, and *CD-8* in gills of infected and non-infected fish.**

## Discussion

Among the world's most productive and widely traded food fish are *O. niloticus* [29]. Of all farmed tilapia, the Nile tilapia is the most cultured freshwater sp. and accounts for around 71% of global tilapia output [30]. In this study, 194 of the 628 Nile tilapia fish samples were infected with myxosporean parasite spores. Nile tilapia was naturally infected with *Myxobolus* sp. (22.5%), where the current results agree with those of Eissa et al. [31], Mohammed et al. [32],

**Table 5. The serum cortisol and glucose levels in control and infected fish.**

| Stressor indicators | Infected fish sp. | Non-infected fish sp. | *P*-value |
|---|---|---|---|
| Cortisol (ng/mL) | 41–62 (51.5 ± 2.3) | 21–30 (25.5 ± 1.8) | 0.000 |
| Glucose (mg/dL) | 95–114 (104.5 ± 3.1) | 38–50 (44 ± 3.1) | 0.001 |

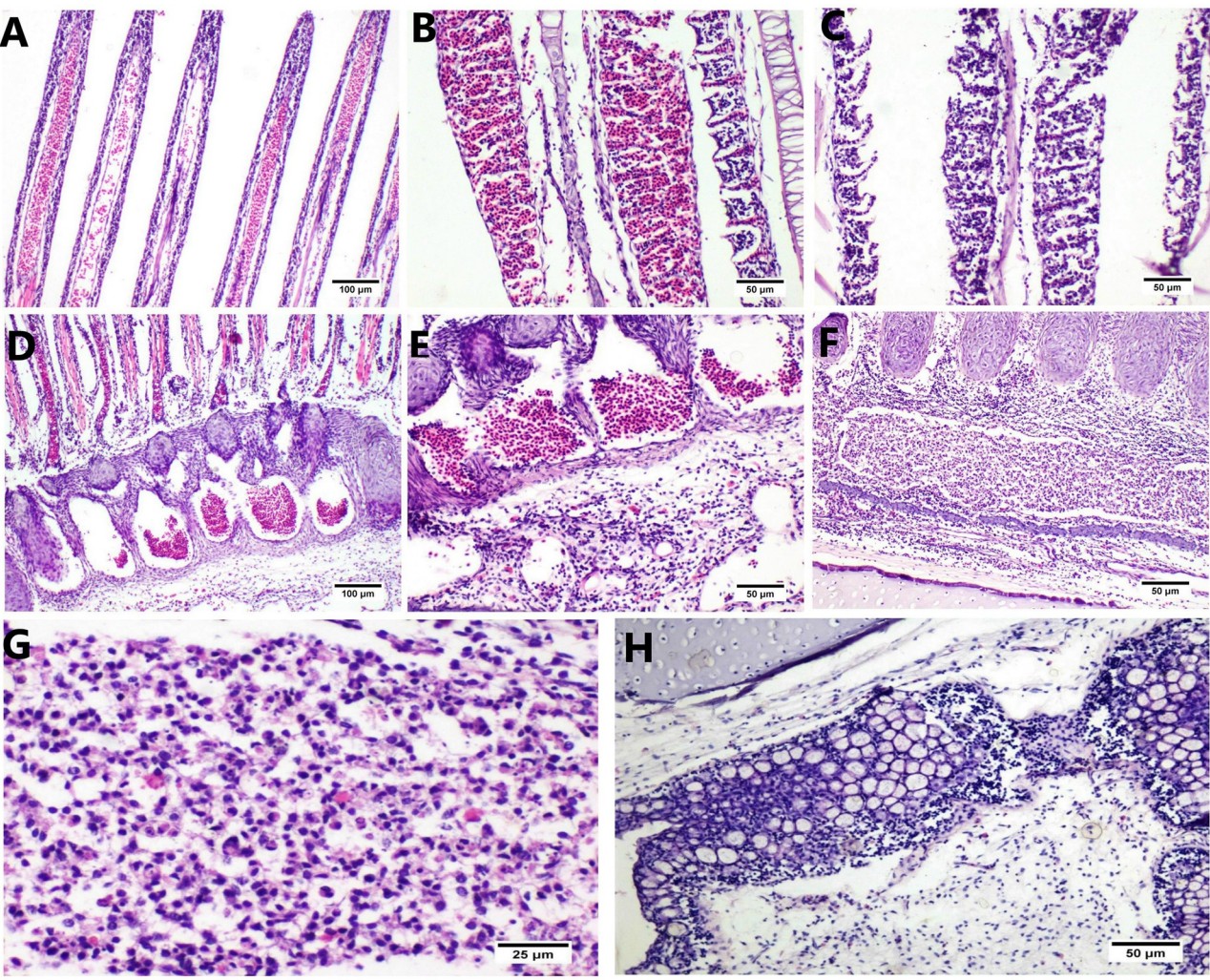

**Fig 6.** (A) Histological section of gills from infected fish stained by H&E showing marked distortion of gill filaments with congestion of the lamellar venous sinuses. (B) Severe hyperemia of capillary sinuses with interlamellar hemorrhages. (C) Extensive fusion of secondary gill lamellae, edema, and sloughing of lamellar epithelium with lymphocytic infiltration. (D) The gill arch shows intense leucocytic infiltration with marked branchial arteries and arteriole hyperemia. (E) Higher magnification of the previous image shows infiltration of lymphocytes mixed with macrophages and EGCs with marked interstitial edema. (F) Intense lymphoplasmacytic infiltration with marked hyperemia of branchial arteries and arterioles. (G) Higher magnification shows infiltration of lymphocytes admixed with macrophages with the sporogenic stage of *Myxobolus* sp. The circulatory path of protozoal dissemination is indicated in the branchial arterial lumen. (H) The gill arch shows hyperplasia of mucous cells comprising the mucosa of the gill arch, with an increased number of basal lymphocytes and lymphocytic infiltration in the submucosa.

and Abdel-Ghaffar et al. [33]. On the other hand, the *Henneguya* sp. prevalence was 8.4%, which agrees with the results of Rabie et al. [34]., El-Seify et al. [35], and Matter et al. [36]. In the current investigation, *Henneguya* sp. and three species of *Myxobolus* were identified. Previous studies [12,31,33,37] compared these species to some of the most morphologically identical species reported.

The availability of new molecular techniques, such as PCR targeting characteristics genes for the myxosporean parasite, supports morphological identification to accurately determine parasitic protozoa [12]. BLAST analysis using the 18S rRNA gene verified the identification of myxosporean parasites, which were categorized into two genera, *Myxobolus* and *Henneguya* sp., and the results were equivalent to those found in the GenBank database. Through pair-

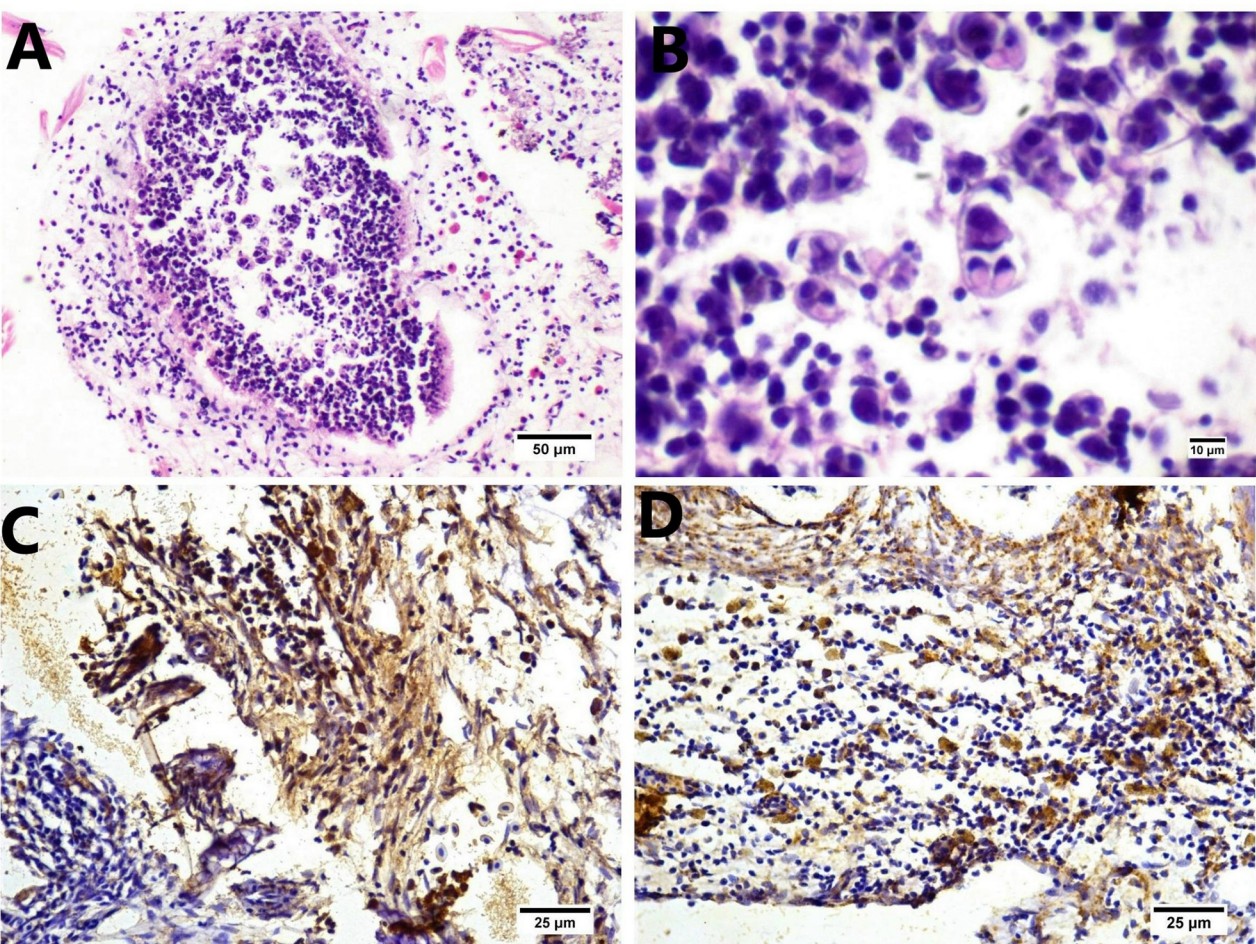

**Fig 7.** (A-B) Histological sections of the gill cavity stained by H&E showing the parasitic cyst surrounded by an inflammatory reaction, mainly lymphocytes and macrophages mixed with eosinophilic granular cells (EGCs). (A) Higher magnification shows an intense aggregation of early sporogenic stages with paired polar capsules of *Myxobolus* sp. mixed with lymphocytes and histiocytes. (B) Immunohistochemistry expression of *iNOS* in smooth muscle cells of the branchial vascular wall, mononuclear cells, and interstitial stroma. (C) The immunohistochemistry expression of *TNF-α* in mononuclear cells infiltrates the interstitial stroma (D).

wise sequence alignment, the identities of *Myxobolus* sp. were confirmed as *M. tilapiae*, *M. agolus*, and *M. brachysporus*. The nucleotide sequence of the 18S rRNA locus of *Henneguya* exhibited similarity with sequences reported by Morsy et al. [38] and Emeish et al. [15] in Egypt, as well as Székely et al. [39] in Hungary and Vieira et al. [40] in Brazil. BLAST analysis of the 18S rRNA gene of *M. agolus*, *M. tilapiae*, and *M. brachysporus* corresponded with findings by Abdel-Gaber et al. [12].

Previous studies indicated that the induction of cellular and humoral host immune responses contributes to establishing host defense mechanisms against parasites [41]. When infectious pathogens, including bacteria, viruses, and parasitic invaders, stimulate fish immune cells, they release inflammatory cytokines. Several soluble and membrane-bound proteins collaborate with the complement system to assist the innate immune system eliminate infections. Jørgensen et al. [42] reported that both bacteria and antibody-antigen (Ag-Ig) complexes can directly trigger complement-mediated mortality.

The present investigation was performed to evaluate several immune genes in the gills of infected and non-infected fish. Although fish gills are frequently contaminated with parasites, the mucus released by these regions is a barrier to other parasites, reducing the total parasite load. As per Dickerson and Findly [43], when a parasitic protozoan infection occurs, *O. niloticus* can mount an immunological response, leading to an upregulation in the expression of numerous genes.

The mucus is the body's first immunological barrier to guard against infections by secreting different lectins, immunoglobulins, complement C-reactive proteins, and lysozymes. According to Zhu et al. [44], these findings suggest the presence of mucous and macrophages in the gills, with a notably higher level of upregulation. However, this accounts for various cytokines secreted by mast cells, macrophages, and lymphocytes.

*IL-1β*, *MHC-II*, *CD-4*, and *CD-8* cytokines can induce fish immunological responses. These genes have received much attention because they are essential in immunological defense and disease management. Compiling foreign peptides from extracellular pathogens and transferring them to helper T cells is a crucial function of these genes. This study unveiled a substantial rise in *IL-1β*, *MHC-II*, *CD-4*, and *CD-8* levels in infected fish tissues, consistent with prior research [9,7,45–49]. However, this may be an immune reaction to reduce parasite infections. In the present study, the cortisol and glucose levels were significantly higher in infected fish than in control non-infected ones in response to parasite infection, such as external protozoans. These results are in agreement with the findings of Marengoni et al. [50], Ellison et al. [51], and Sandoval-Gío et al. [52].

The histopathological findings of tilapia's gills showed marked histopathological alterations that extended from the gill rackers, arch, and filaments. These alterations correlated with the direct injury induced by the existence of different stages of *Myxobolus* species in gill structures. The *Myxobolus* cyst was detected only in the gill arch. In contrast, the gill filaments showed edema, lamellar congestion, and hemorrhages directly related to *Myxobolus* species in branchial arteries and arteriole with severe inflammatory reactions, with subsequent induction of oxidative stress and tissue damage. Similar findings were described by Hardiono and Yanuhar [53], who observed gill damage induced by direct irritation of the gill epithelium by myxosporean. The existence of marked inflammatory reactions in gill structures was detected mainly in macrophages and lymphocytes with few EGC infiltration. These marked inflammatory reactions were associated with increased expression of *iNOS* and *TNF-α* in gill structures; however, macrophages produce nitric oxide induced by *iNOS* release after *Myxobolus* infection of susceptible trout lodge strains [54]. Furthermore, there was upregulation of *iNOS* expression, which was detected from 1 to 8 days post-exposure. In addition, Akram et al. [55] stated that the elevated *iNOS* expression in the tissues of Nile tilapia infected by myxosporeans was correlated with the inflammatory response and tissue damage.

## Conclusion

Myxosporeans are among the fish parasites that affect fish productivity, as they impact fish growth and reproduction, resulting in large fish deaths in farms and hatcheries. This study has been focused on morpho-molecular identification for the myxosporean parasites infecting Nile tilapia from three governorates in Egypt and assessment of gene expression of different cytokines (*IL1-β*, *MHC-II*, *CD-4*, and *CD-8*) in tissues. Additionally, a correlation was performed between the developed histopathological alterations and inflammatory reactions in gills and the immunohistochemical expression of *iNOS* and *TNF-α*. The present study provides new insights into oxidative stress biomarkers in Nile tilapia infected with Myxosporeans

and elucidates the gill's immune status changes as a portal of entry for protozoa that contribute to tissue damage.

## Author Contributions

**Data curation:** Reem M. Ramadan, Olfat A. Mahdy, Mai A. Salem.

**Formal analysis:** Reem M. Ramadan, Mai A. Salem.

**Funding acquisition:** Faten F. Mohammed.

**Investigation:** Reem M. Ramadan, Mohamed A. El-Saied, Faten F. Mohammed, Mai A. Salem.

**Methodology:** Reem M. Ramadan, Mohamed A. El-Saied.

**Supervision:** Olfat A. Mahdy.

**Writing – original draft:** Reem M. Ramadan, Mai A. Salem.

**Writing – review & editing:** Olfat A. Mahdy, Mohamed A. El-Saied, Faten F. Mohammed.

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
