## [Decision Letter · Decision Letter 0]

25 Mar 2024

PONE-D-24-07797Novel Insights into Immune Stress Markers Associated with Myxosporeans Gill Infection in Nile Tilapia (Molecular And Immunohistochemical Studies)PLOS ONE

Dear Dr. Mohammed,

Thank you for submitting your manuscript to PLOS ONE. After careful consideration, we feel that it has merit but does not fully meet PLOS ONE’s publication criteria as it currently stands. Therefore, we invite you to submit a revised version of the manuscript that addresses the points raised during the review process.

We look forward to receiving your revised manuscript.

Kind regards,

Ishtiyaq Ahmad, Ph.D

Academic Editor

PLOS ONE

Journal Requirements:

4. We suggest you thoroughly copyedit your manuscript for language usage, spelling, and grammar. If you do not know anyone who can help you do this, you may wish to consider employing a professional scientific editing service. 

5. To comply with PLOS ONE submissions requirements, in your Methods section, please provide additional information regarding the experiments involving animals and ensure you have included details on (1) methods of sacrifice, (2) methods of anesthesia and/or analgesia, and (3) efforts to alleviate suffering.

6. Thank you for stating in your Funding Statement: 

"The authors acknowledge the Deanship of Scientific Research, Vice Presidency for Graduate Studies and Scientific Research at King Faisal University, Saudi Arabia, for financial support under the annual funding track (GRANT5,913)"

7. Thank you for stating the following in the Acknowledgments Section of your manuscript: 

"The authors acknowledge the Deanship of Scientific Research, Vice Presidency for Graduate Studies and Scientific Research at King Faisal University, Saudi Arabia, for financial support under the annual funding track  GRANT5,913"

"The authors acknowledge the Deanship of Scientific Research, Vice Presidency for Graduate Studies and Scientific Research at King Faisal University, Saudi Arabia, for financial support under the annual funding track (GRANT5,913)"

Reviewers' comments:

Reviewer's Responses to Questions

**Comments to the Author**

1. Is the manuscript technically sound, and do the data support the conclusions?

Reviewer #1: Yes

Reviewer #2: Yes

2. Has the statistical analysis been performed appropriately and rigorously? 

Reviewer #1: Yes

Reviewer #2: Yes

3. Have the authors made all data underlying the findings in their manuscript fully available?

Reviewer #1: Yes

Reviewer #2: Yes

4. Is the manuscript presented in an intelligible fashion and written in standard English?

Reviewer #1: Yes

Reviewer #2: Yes

5. Review Comments to the Author

Reviewer #1: The study conducted by Ramadan et al. is very interesting and will be useful for PLOS one readers. The manuscript the hypothesis of the study is clear and the manuscript is well-written. Minor issues are needed for improving the manuscript before being published.

Abstract

1-the abstract should be informative, all genes abbreviations should be mentioned in full formula for the first time, then use the abbreviated form

2- the authors should mention the methodology in brief (I mean, no of fish used in the study, no of groups, no of replicates, duration of the study).

3- The abbreviated form of genes should be italic

4-line 69, the abbreviations should be written in full formula

5-line 71, the abbreviated gene name should be italic

6-line 98, Oreochromis niloticus should replace by O. niloticus

7-line 153, µl should be replaced by µL and should be revised in all manuscript.

8-line 179, remove (,)

9-line 182, add the name of anti-coagulant

10-190, which organs were sampled and no of samples

11-line 202, what about normality test?

Reviewer #2: Dear Editor

I would like to thank you for giving me this opportunity to review this manuscript.

This manuscript aims to assess the stress response of Nile Tilapia to the infection by Myxosporeans via evaluating immune response at different levels, protein changes, hormonal and tissue histopathological and immunohistochemical alterations with good correlation of the results and interpretation of data based on the obtained results. This is an interesting study however the following points need to be addressed:

The title of the manuscript is very novel and defines the status of the article. All figures are beneficial in illustration of the results.

The authors mentioned that blood samples from five infected fish and five uninfected fish were subjected to biochemical analysis, so the authors before this step were examined the fish to determine whether they infected or not, so the handling of fish is considered a stressor that may alter the level of serum cortisol. So, you must highlight light after which period from handling you collect the blood samples?

line 183, the authors mentioned that: had blood drawn from their caudal veins. To extract serum by biochemical analysis,), please this sentence grammatically.

line 184, the authors mentioned that: the sample was then frozen at −20 °C.), you should mention that this was a serum sample.

In the histopathological section, why did the authors select TNF-α and iNOS to perform tissue expression?

the authors should mention the software used for image capturing and scale bar application.

the authors should clarify the significance in table (5).

6. PLOS authors have the option to publish the peer review history of their article (what does this mean?). If published, this will include your full peer review and any attached files.

Reviewer #1: No

Reviewer #2: No

---

## [Author Response · Author response to Decision Letter 0]

15 Apr 2024

Dear Editor,

Thank you for your great efforts in reviewing our manuscript No. PONE-D-24-07797, Entitled: (Novel Insights into Immune Stress Markers Associated with Myxosporeans Gill Infection in Nile Tilapia (Molecular And Immunohistochemical Studies). We responded to all the reviewer’s requirements and observations inside the text and identified the response point by point in the attached table. All corrections are highlighted and changed in the main text. 

We appreciate all your efforts, thank you again. Do not hesitate to ask us about any other observations or comments.

Thank you

The manuscript Authors.

List our responses on different points that were corrected inside the text and mentioned here point by point in the present table:

Journal Requirements & Reviewer Comments Response

Journal Requirements

1-Please ensure that your manuscript meets PLOS ONE's style requirements, including those for file naming. Done.

2-Please include your tables as part of your main manuscript and remove the individual files. Done.

3-In your Methods section, please provide additional information regarding the permits you obtained for the work. Please ensure you have included the full name of the authority that approved the field site access Done.

4-We suggest you thoroughly copyedit your manuscript for language usage, spelling, and grammar. If you do not know anyone who can help you do this, you may wish to consider employing a professional scientific editing service. We have edited the manuscript for language and grammar, from the aspect of fluency.

Please find the attached Editing Certificate at the end of the table.

5-please provide additional information regarding the experiments involving animals and ensure you have included details on (1) methods of sacrifice, (2) methods of anesthesia and/or analgesia, and (3) efforts to alleviate suffering. Fish are anesthetized by immersing them in an anesthetic bath containing 150 mg/L of tricaine methanesulfonate (MS-222 for fish) so that the drug is absorbed through the gills and rapidly enters the bloodstream. Fish are then euthanized under anesthesia to alleviate fish suffering. Fish are euthanized by decapitation.

6-Thank you for stating in your Funding Statement:

 Done, a detailed Financial Disclosure Statement was added to the cover letter

7-Thank you for stating in the Acknowledgments Section of your manuscript

"The authors acknowledge the Deanship of Scientific Research, Vice Presidency for Graduate Studies and Scientific Research at King Faisal University, Saudi Arabia, for financial support under the annual funding track (GRANT5,913)"

Please include your amended statements within your cover letter; we will change the online submission form on your behalf. Done 

8- Please review your reference list to ensure that it is complete and correct. If you have cited papers that have been retracted, please include the rationale for doing so in the manuscript text, or remove these references and replace them with relevant current references. Any changes to the reference list should be mentioned in the rebuttal letter that accompanies your revised manuscript. If you need to cite a retracted article, indicate the article’s retracted status in the References list and also include a citation and full reference for the retraction notice. We added 4 missed references that were cited in tables that were previously separated from the entire manuscript but when we included the tables within the manuscript the references were cited in order, theses references were:

1-Heinecke, R. D., & Buchmann, K. (2013). Inflammatory response of rainbow trout Oncorhynchus mykiss (Walbaum, 1792) larvae against Ichthyophthirius multifiliis. Fish & shellfish immunology, 34(2), 521-528.‏

2-Suprapto R, Alimuddin N, S., Imron, Marnis, H., Iswanto, Bambang, (2017) MHC-II marker potential linked to motile aeromonad septicaemia disease resistance in african catfish (clarias gariepinus). Indonesian Aquaculture Journal 12(1):21–28

3-Raida MK, Buchmann K. (2008) Development of adaptive immunity in rainbow trout, Oncorhynchus mykiss (Walbaum) surviving an infection with Yersinia ruckeri. Fish & Shellfish Immunology.

4-Raida, M.K. & Buchmann, K. (2007) Temperature-dependent expression of immune-relevant genes in rainbow trout following Yersinia ruckeri vaccination. Diseases of aquatic organisms,77, 41-52.

Additionally, one reference was already cited in the discussion section and also cited in tables so its order was changed, this reference is Emeish, W. F., Fawaz, M. M., Al-Amgad, Z., & Hussein, N. M. (2022). Henneguya species infecting the gastrointestinal tract of Clarias gariepinus from the Nile River. Diseases of Aquatic Organisms, 148, 43-56.‏

It was cited in the discussion section by number 34 but after adding it the tables were cited as number 15

Reviewer Comments

Reviewer 1 

1-The abstract should be informative, all genes abbreviations should be mentioned in full formula for the first time, then use the abbreviated form Done.

2-The authors should mention the methodology in brief (I mean, no of fish used in the study, no of groups, no of replicates, duration of the study). Done.

3-The abbreviated form of genes should be italic Done.

4-line 69, the abbreviations should be written in full formula Done.

5-line 71, the abbreviated gene name should be italic Done.

6-line 98, Oreochromis niloticus should replace by O. niloticus Done.

7-line 153, µl should be replaced by µL and should be revised in all manuscript. Done to all of them.

8-line 179, remove (,) Removed.

9-line 182, add the name of anti-coagulant Added in the manuscript.

10-190, which organs were sampled and no of samples Mentioned in the methodology section 

11-line 202, what about normality test? The normality test was conducted using SPSS software ANOVA analysis.

Reviewer 2 

The authors mentioned that blood samples from five infected fish and five uninfected fish were subjected to biochemical analysis, so the authors before this step were examined the fish to determine whether they infected or not, so the handling of fish is considered a stressor that may alter the level of serum cortisol. So, you must highlight light after which period from handling you collect the blood samples? Fish are easily stressed by handling and stress can result in immunosuppression. Anesthetics are used before blood sampling to prevent any stressors. 

line 183, the authors mentioned that: had blood drawn from their caudal veins. To extract serum by biochemical analysis,), please this sentence grammatically. Done.

line 184, the authors mentioned that: the sample was then frozen at −20 °C.), you should mention that this was a serum sample. Done.

In the histopathological section, why did the authors select TNF-α and iNOS to perform tissue expression? TNF-α was selected as it is considered a marker for inflammation and the intensity of expression in tissue was exaggerated in the intense inflammatory reaction, while iNOS was selected as it expressed the grade of oxidative stress exerted in tissue associated with the inflammatory reaction

the authors should mention the software used for image capturing and scale bar application. Done, is mentioned in the methodology section

the authors should clarify the significance in table (5). Clarified and added in the manuscript.

---

## [Decision Letter · Decision Letter 1]

30 Apr 2024

Novel Insights into Immune Stress Markers Associated with Myxosporeans Gill Infection in Nile Tilapia (Molecular And Immunohistochemical Studies)

PONE-D-24-07797R1

Dear Dr. Mohammed,

We’re pleased to inform you that your manuscript has been judged scientifically suitable for publication and will be formally accepted for publication once it meets all outstanding technical requirements.

Kind regards,

Ishtiyaq Ahmad, Ph.D

Academic Editor

PLOS ONE

Reviewers' comments:

Reviewer's Responses to Questions

**Comments to the Author**

1. If the authors have adequately addressed your comments raised in a previous round of review and you feel that this manuscript is now acceptable for publication, you may indicate that here to bypass the “Comments to the Author” section, enter your conflict of interest statement in the “Confidential to Editor” section, and submit your "Accept" recommendation.

Reviewer #1: All comments have been addressed

Reviewer #2: All comments have been addressed

2. Is the manuscript technically sound, and do the data support the conclusions?

Reviewer #1: Yes

Reviewer #2: Yes

3. Has the statistical analysis been performed appropriately and rigorously? 

Reviewer #1: Yes

Reviewer #2: Yes

4. Have the authors made all data underlying the findings in their manuscript fully available?

Reviewer #1: Yes

Reviewer #2: Yes

5. Is the manuscript presented in an intelligible fashion and written in standard English?

Reviewer #1: Yes

Reviewer #2: Yes

6. Review Comments to the Author

Reviewer #1: Congratulation. All comments have been addressed by authors. No additional comments are required. Now the manuscript is suitable for publication

Reviewer #2: All my inquiries have been thoroughly addressed by the authors of this manuscript and consequently I accept it for publication

7. PLOS authors have the option to publish the peer review history of their article (what does this mean?). If published, this will include your full peer review and any attached files.

Reviewer #1: No

Reviewer #2: No

---

## [Editor Report · Acceptance letter]

14 May 2024

PONE-D-24-07797R1 

PLOS ONE

Dear Dr. Mohammed, 

I'm pleased to inform you that your manuscript has been deemed suitable for publication in PLOS ONE. Congratulations! Your manuscript is now being handed over to our production team.

Kind regards, 

on behalf of

Dr. Ishtiyaq Ahmad 

Academic Editor

PLOS ONE